# MetaBox-v2: A Unified Benchmark Platform for Meta-Black-Box Optimization

**Zeyuan Ma**[1], **Yue-Jiao Gong**[1,*], **Hongshu Guo**[1], **Wenjie Qiu**[1], **Sijie Ma**[1],
**Hongqiao Lian**[1], **Jiajun Zhan**[1], **Kaixu Chen**[1], **Chen Wang**[1], **Zhiyang Huang**[1],
**Zechuan Huang**[1], **Guojun Peng**[1], **Ran Cheng**[2], **Yining Ma**[3]
[1]South China University of Technology  [2]Hong Kong Polytechnic University
[3]Massachusetts Institute of Technology

## Abstract

Meta-Black-Box Optimization (MetaBBO) streamlines the automation of optimization algorithm design through meta-learning. It typically employs a bi-level structure: the meta-level policy undergoes meta-training to reduce the manual effort required in developing algorithms for low-level optimization tasks. The original MetaBox (2023) provided the first open-source framework for reinforcement learning-based single-objective MetaBBO. However, its relatively narrow scope no longer keep pace with the swift advancement in this field. In this paper, we introduce MetaBox-v2 (https://github.com/MetaEvo/MetaBox) as a milestone upgrade with four novel features: 1) a unified architecture supporting RL, evolutionary, and gradient-based approaches, by which we reproduce 23 up-to-date baselines; 2) efficient parallelization schemes, which reduce the training/testing time by $10-40$x; 3) a comprehensive benchmark suite of 18 synthetic/realistic tasks (1900+ instances) spanning single-objective, multi-objective, multi-model, and multi-task optimization scenarios; 4) plentiful and extensible interfaces for custom analysis/visualization and integrating to external optimization tools/benchmarks. To show the utility of MetaBox-v2, we carry out a systematic case study that evaluates the built-in baselines in terms of the optimization performance, generalization ability and learning efficiency. Valuable insights are concluded from thorough and detailed analysis for practitioners and those new to the field.

## 1 Introduction

Black-Box-Optimization (BBO) represents challenging optimization tasks in practice. For decades, many BBO optimizers [1–4] are developed and widely discussed. A key limitation of traditional BBO optimizers is that they require human experts to design effective algorithms, which might result in design bias to adapt for novel optimization scenarios [5]. To address this, recent Meta-Black-Box Optimization (MetaBBO) [6, 7] researches propose meta-learning algorithm design policy by a bi-level framework: the meta-level policy is trained on a problem distribution to maximize the performance of low-level BBO optimizer. The trained policy is expected to generalize on unseen problems. Considering MetaBBO lacks decent benchmark, MetaBox [8] in 2023 served as the first benchmark platform for developing and evaluating MetaBBO approaches. In particular, this original version focused on a specific optimization scenario: single-objective optimization, and a specific learning paradigm: MetaBBO with reinforcement learning (MetaBBO-RL). The reason behind is that before 2023, a primary research focus in MetaBBO was exploring how to incorporate RL [9] with BBO optimizers. With 8 popular MetaBBO-RL baselines and 3 basic testsuites, MetaBox provides

---

*Yue-Jiao Gong is the corresponding author (gongyuejiao@gmail.com).

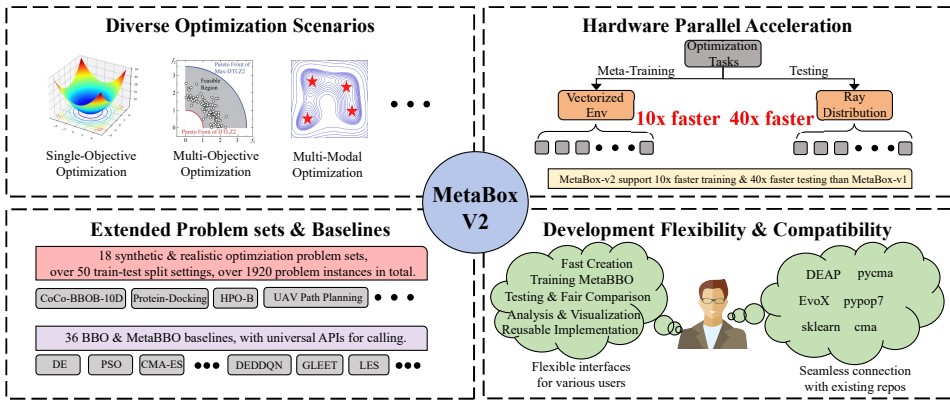

Figure 1: The four novel and user-friendly features of MetaBox-v2.

fully automatic train-test-log workflow with minimal development requirement. It has received considerable recognition in the field and gathered a MATLAB extension recently [10].

Promising as it is, MetaBBO researches grow up rapidly within the last two years. On the one hand, more and more novel ideas came out in terms of flexible learning paradigms. According to the latest survey [6], four major learning paradigms have been discussed: MetaBBO-RL, MetaBBO-SL [11–13] where supervised learning is used to train the meta-level policy, MetaBBO-NE [14, 15] where neuroevolution [16] is adopted, and MetaBBO-ICL [17–19] where LLMs with in-context learning [20] serve as meta-level policies for algorithm design. On the other hand, MetaBBO's potential has been widely explored in diverse optimization fields such as multi-objective optimization [21], multi-modal optimization [22, 23], large-scale global optimization [24, 25], multi-task optimization [26], etc. The original MetaBox no longer keeps pace with the swift advancement in the field.

We therefore propose MetaBox-v2 through fundamental improvements that systematically address the above limitations while inheriting the benefits of the original Metabox. To summarize, this study presents the following key contributions to advancing the MetaBBO benchmark research:

**1. Milestone Framework Upgrade (MetaBox-v2):** The upgraded MetaBox-v2 introduces four synergistic enhancements through framework innovations, as illustrated in Figure 1.

*1) Unified Integration of All Four MetaBBO Paradigms:* Through redefine the algorithmic interfaces, we now propose the *MetaBBO Template* for algorithm development. It serves as the first framework capable of supporting all the four distinct MetaBBO paradigms: MetaBBO-RL, MetaBBO-SL, MetaBBO-NE, and MetaBBO-ICL. Based on the unified framework, we also extend our baseline library from 8 to 23 algorithms.

*2) Efficient Parallelization:* MetaBBO approaches are typically time-consuming due to the nested bi-level structure. In MetaBox-v2, we introduce two parallel schemes: vectorized optimization environment and instance-level distributed evaluation, to accelerate MetaBBO by $10 - 40$x.

*3) Rich Benchmarks:* To include diverse optimization problem types, we rewrite the *Problem* class as an inheritable class. By inheriting from *Problem*, complex optimization problems such as multi-objective and multi-task ones can be seamlessly integrated, allowing polymorphism in different problem-specific behavior. MetaBox-v2 extend the testsuites from 3 to 18 synthetic/realistic tasks.

*4) Plentiful and Extensible Interfaces:* We thoroughly upgrade MetaBox's developer flexibility. Every detailed process data are systematically recorded as metadata, which could be used for customized analysis by users. Furthermore, to match the open-source ecosystem, MetaBox-v2 provides plentiful interfaces to external resources. We prepare a systematic online documentation to guide the users.

**2. Comprehensive Benchmarking Study:** A comprehensive benchmarking study is conducted to showcase the practical value of MetaBox-v2, where up-to-date MetaBBO baselines are fairly trained and evaluated in terms of their performance, efficiency, generalization ability, etc. Our analysis yields valuable insights, particularly the significant variance in baseline generalization across testsuites and the critical trade-offs between learning efficiency and performance robustness. These findings establish empirical guidelines for practitioners while identifying promising research directions, accelerating future advancements in MetaBBO algorithm development.

Table 1: Comparison to related benchmarks. *#Optimization Scopes*: supported optimization problem types; *Learning Support*: supported MetaBBO learning paradigms; *Parallel*: hardware-level parallelism support; *#Problem*: the number of problem instances (*#synthetic + #realistic*); *#MetaBBO Baseline*: the number of MetaBBO baselines; *Template*: Template coding support; *Auto*: automated train/test workflow; *Custom*: configurable settings; *Visual*: visualization tools support; *Compatibility*: compatibility with open-source resources.

| | #Optimization Scopes | Learning Support | Parallel | #Problems | #MetaBBO Baselines | Template | Auto | Custom | Visual | Compatibility |
|---|---|---|---|---|---|---|---|---|---|---|
| COCO [42] | 4 | × | × | 481+0 | × | ✓ | ✓ | × | ✓ | few |
| CEC [43] | 1 | × | × | 30+0 | × | × | × | × | × | none |
| IOHprofiler [44] | 3 | × | × | 55+0 | × | ✓ | × | ✓ | ✓ | few |
| Bayesmark [45] | 1 | × | × | 0+228 | × | ✓ | ✓ | × | × | few |
| Zigzag [46] | 1 | × | × | 4+0 | × | × | × | ✓ | × | none |
| Engineering [47] | 1 | × | × | +57 | × | × | × | × | × | none |
| MA-BBOB [48] | 1 | × | × | 1000+0 | × | × | × | × | × | none |
| BBOPlace [49] | 1 | × | × | 0+14 | × | × | ✓ | × | × | none |
| PyPop7 [50] | 2 | × | × | 92+11 | × | × | × | × | × | few |
| EvoX [51] | 2 | × | ✓ | 44+50 | × | ✓ | × | ✓ | ✓ | rich |
| MetaBox [8] | 1 | RL | × | 54+280 | 8 | ✓ | ✓ | ✓ | ✓ | few |
| MetaBox-v2 | 5 | RL,SL, NE,ICL | ✓ | 541+1393 | 23 | ✓ | ✓ | ✓ | ✓ | rich |

## 2 Related Works

**MetaBBO.** We first illustrate the bi-level paradigm of Meta-Black-Box-Optimization (MetaBBO) [6] in Figure 2. In low-level optimization environment, a BBO optimizer $\mathcal{A}$ is maintained to optimize a problem $p$ sampled from distribution $\mathcal{P}$. At each optimization step $t$, optimization status features are extracted from the current optimization process (such as population and objective values information). Then in meta level, an algorithm design policy $\pi_\theta$ (with learnable parameters $\theta$) outputs a desired design $\omega_i^t$ by $\omega_i^t = \pi_\theta(s_i^t)$. $\mathcal{A}$ optimizes $p$ by $\omega_i^t$ for one step. A performance measurement function $r_t$ is used to evaluate the performance gain obtained by this algorithm design decision. Suppose $T$ optimization steps are allowed for the low-level optimization process, then $\pi_\theta$ is meta-trained to maximize a meta-objective formulated as: $J(\theta) = \mathbb{E}_{p \in \mathcal{P}}[\sum_{t=1}^{T} r_t]$, which is expectation of accumulated single step performance gain

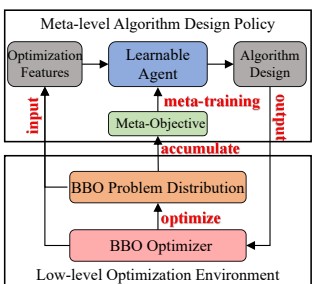

Figure 2: Bi-level Paradigm of existing MetaBBOs.

over all problem instances in $\mathcal{P}$. In practice, a training problem set serves as the distribution $\mathcal{P}$. It is important to note that in this brief introduction we use $\omega$ as an abstract notation for algorithm design, considering that MetaBBO provides a universal concept framework for diverse algorithm design tasks such as algorithm selection, algorithm configuration etc. $\omega$ is hence a flexible and abstract concept to represent specific design choice under the given context. For a comprehensive summary of different algorithm design categories, we refer to MetaBBO survey [6] for further reading.

Following this paradigm, a wide array of MetaBBO approaches have been proposed, which further diverge into four same-end branches according to the learning techniques they adopt [6, 7]. The four branches are: 1) MetaBBO-RL: those first model the algorithm design process as a Markov Decision Process (MDP), then employ effective RL techniques to learn well-performing policies. Initial works such as DEDDQN [27], LDE [28], DEDQN [29] and RLEPSO [30] focus on designing dynamic configuration strategy for BBO optimizer. Following them, in-depth exploration include high-capacity neural policy [31, 32], complete optimizer generation [33, 34], optimization feature learning [35, 36] and efficient offline learning [37, 38]; 2) MetaBBO-SL: this branch originates from RNN-opt [11], where given a solution as input, a RNN is used to auto-regressively iterate it for better solution. The RNN is trained by minimizing the differentiable objective function. Although this paradigm requires white-box (differentiable) problems for training, recent works such as GLHF [13] and B2Opt [12] demonstrate that only training on synthetic problems is sufficient for generalization towards unseen problems; 3) MetaBBO-NE: where a the meta-level policy is learned by evolutionary optimization on its net parameters [14, 15, 39]; 4) MetaBBO-ICL: where a general LLM serves as either the low-level optimizer [17–19], or a meta-level configuration policy [40, 41]. The textual optimization process information is regarded as the context and learned by the LLM to propose algorithm designs. The fast development in MetaBBO field anticipates corresponding benchmarks.

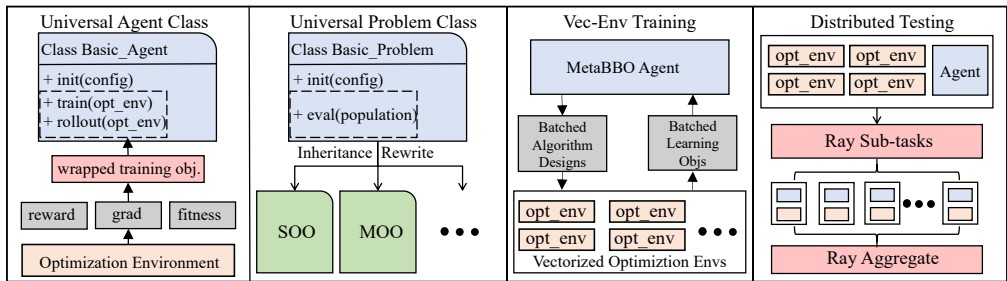

Figure 3: Major architecture adjustments in MetaBox-v2.

**Related Benchmarks.** A comparison of MetaBox-v2 to representative and up-to-date BBO benchmarks is presented in Table 1 to show the novelty of our work. Apart from those have been reviewed and compared in MetaBox [8], latest efforts on developing BBO benchmark include: 1) Engineering [47]: a collection of real world engineering optimization problems such as heat exchanger network design, industrial chemical process optimization, etc; 2) MA-BBOB [48]: a many-affine problem sets constructed from COCO [42] by interpolation operations on COCO's problem instances, resulting in diversified synthetic instances; 3) BBOPlace [49]: chip placement tasks which represents complex and challenging optimization scenarios; 4) PyPop7 [50]: a comprehensive benchmark platform featured by massive number of BBO optimizers, which include decades of advanced optimizers with different types and specialized scenarios. 5) EvoX [51]: a high-efficiency benchmark platform featured by its distributed GPU-accelerated evaluation framework, with over 100x speedups than traditional BBO benchmarks such as CoCo and PyPop7. Notably, MetaBox, across its two versions, remains the sole platform that supports MetaBBO's bi-level framework to streamline the development and benchmarking processes in this research domain.

## 3 MetaBox-v2

### 3.1 Unified MetaBBO Interface

**Compatibility with Diverse MetaBBO.** As reviewed in Section 2, the rapid development in MetaBBO has witnessed the exploration of various learning paradigms. A fundamental challenge is that, while sharing the bi-level paradigm, their underlying learning forms differ with each other. MetaBBO-RL is built on MDP, necessitating a reward signal from the low-level optimization environment to train the meta-level RL agent through trial-and-error. MetaBBO-SL and MetaBBO-NE require gradient information and fitness-like feedback, respectively. To achieve this, in MetaBox-v2, we replace the original RL-specific agent class with a unified *Basic_Agent* class featuring universal train and rollout interfaces. This is achieved through a wrapper function to transform different learning objective forms into a universal data object (shown in the left of Figure 3).

By such a novel design, we reproduce 15 more representative MetaBBO baselines upon the original MetaBox, including **1) MetaBBO-RL**: NRLPSO [52], RLDAS [32], SYMBOL [33], GLEET [31], RLDEAFL [36], Surr_RLDE [37], MADAC [21], PSORLNS [53], RLEMMO [22], L2T [26]; **2) MetaBBO-SL**: GLHF [13], B2OPT [12]; **3) MetaBBO-NE**: LES [15], LGA [14]; **4) MetaBBO-ICL**: OPRO [17]. In summary, MetaBox-v2 now supports 36 baselines including 23 MetaBBO baselines and 13 traditional BBO baselines. It is capable of providing not only comprehensive comparisons and analysis usages, but also a formal tutorial for those new to this field.

**Scalable Testsuites Library.** While the original MetaBox supported three synthetic/realistic testsuites for single-objective optimization (SOO), the up-to-date MetaBBO approaches have been initiated to multi-objective optimization (MOO) [21, 54], large-scale global optimization (LSGO) [24, 25], multi-modal optimization (MMO) [22] and multi-task optimization (MTO) [26, 55]. To keep pace with MetaBBO's advancement so as to embrace users from diverse optimization sub-domains, we make a key adjustment to generalizing the SOO-specific problem class in MetaBox into a polymorphic *Basic_Problem* parent class (shown in the second column of Figure 3). This abstract base class with its core eval() interface enables problem specialization through inheritance, namely, users implement domain-specific evaluation logic by overriding this method.

Table 2: Diverse BBO testsuites in MetaBox-v2.

| Name | Type | Dimension | maxFEs | Size | Scenario | Description | License |
|------|------|-----------|--------|------|----------|-------------|---------|
| *bbob-10D*[42] | SOO | 10D | 2E4 | 24 | synthetic | Single-objective instances in CoCo | BSD-3-Clause |
| *bbob-30D*[42] | SOO | 30D | 5E4 | 24 | synthetic | Single-objective instances in CoCo | BSD-3-Clause |
| *bbob-noisy-10D*[42] | SOO | 10D | 2E4 | 24 | synthetic | *bbob-10D* with gaussian noise | BSD-3-Clause |
| *bbob-noisy-30D*[42] | SOO | 30D | 5E4 | 24 | synthetic | *bbob-30D* with gaussian noise | BSD-3-Clause |
| *hpo-b*[63] | SOO | 2-16D | 2E3 | 935 | realistic | Hyper-parameter optimization | MIT License |
| *uav*[65] | SOO | 30D | 2.5E3 | 56 | realistic | UAV path planning tasks | Attribution 4.0 |
| *protein*[64] | SOO | 12D | 2E3 | 280 | realistic | Simplified protein-docking instances | Attribution 4.0 |
| *lsgo*[60] | LSGO | ≥905D | 3E6 | 20 | synthetic | Large-scale problem instances | GPL-3.0 |
| *ne*[51] | LSGO | ≥1000D | 2.5E3 | 66 | realistic | Neuroevolution for control tasks | GPL-3.0 |
| *zdt*[56] | MOO | 10-30D | 5E3 | 5 | synthetic | A group of bi-objective problems | Apache-2.0 |
| *uf*[58] | MOO | 30D | 5E3 | 10 | synthetic | Multi-objective problem instances | Apache-2.0 |
| *dtlz*[57] | MOO | 6-29D | 5E3 | 46 | synthetic | Scalable multi-objective problems | Apache-2.0 |
| *wfg*[59] | MOO | 12-38D | 5E3 | 117 | synthetic | Complex multi-objective problems | Apache-2.0 |
| *moo-uav*[56] | MOO | 30D | 2.5E3 | 56 | realistic | Multi-objective form of *uav* | Apache-2.0 |
| *mmo*[61] | MMO | 1-20D | 5E4-4E5 | 20 | synthetic | Standard multi-modal problems | Simplified BSD |
| *cec2017mto*[62] | MTO | 25-50D | 2.5E4 | 9 | synthetic | Multi-task problems in CEC2017 | - |
| *wcci2020*[62] | MTO | 50D | 6.25E5 | 10 | synthetic | Multi-task problems in WCCI2020 | - |
| *wcci2020-aug* | MTO | 50D | 1.25E5 | 127 | synthetic | Flexible combinations of *wcci2020* | - |

Specifically, we have integrated 18 testsuites with over 1900 problem instances from diverse problem types into MetaBox-v2. These problems include not only representative synthetic benchmark functions for SOO [42], MOO [56–59], LSGO [60], MMO [61] and MTO [62], but also realistic testsuites with challenging optimization characteristics widely collected from AutoML [63], protein science [64], UAV system [65] and robotics [51]. We present basic information of them in Table 2. More details are accessible at the online documentation.

**Open-Source Ecosystem.** MetaBox-v2 exemplifies exceptional extensibility through strategic integration with established optimization frameworks. For example, some built-in BBO baselines are implemented by calling powerful platforms such as DEAP [66], PyCMA [67], PyPop7 [50] etc. Some testsuites are borrowed from emerging benchmark platforms such as EvoX [51] to further acquire in-testing acceleration. We provide point-to-point tutorial documentation to connect users with these flexible usages.

## 3.2 Efficiency Optimization

**Parallel Training.** The time-consuming training caused by MetaBBO's bi-level nested paradigm is a critical but understudied bottleneck in current literature. Our preliminary experiments reveal that serialized environment evaluations in the original MetaBox lead to prohibitive training times when handling modern testsuite scales, posing barriers to the rapid development of MetaBBO field. In MetaBox-v2, to address this efficiency issue, we propose a novel parallel scheme termed as *vectorized optimization environment* to accelerate MetaBBO's training. In specific, as illustrated in the third column of Figure 3, during the training, we simultaneously construct a batch of low-level optimization environments and wrap them into a vectorized environment based on Tianshou [68]. Then the meta-level agent could perform batched algorithm designs via multi-processing parallelization in the vectorized environment. This allows parallel collection of learning signals (rewards/gradients/fitness measures) across multiple environments and problem instances, which are aggregated into mini-batches for efficient meta-policy updates. To the best of our knowledge, MetaBox-v2 is the first development examplar to make MetaBBO's training parallel.

**Parallel Testing.** MetaBox-v2 provides Ray-based parallel scheme [69] for distributed testing of MetaBBO/BBO baselines. As illustrated in the right of Figure 3, given a MetaBBO's meta-level agent and a testsuite, we first copy the agent for each testing run and use Ray to construct the corresponding sub-tasks. Then all sub-tasks are distributed into independent CPU/GPU cores for parallel testing. The testing results in these sub-tasks are aggregated automatically by Ray's handler. To show the detail, we first assume a testing scenario where $N$ problem instances compose the testing set, $B$ baselines to be tested, and $R$ independent runs to ensure the statistical robustness. Naive testing procedure requires nested 3-layer loop to complete all these $N \times B \times R$ testing runs. In contrast, in MetaBox-v2, we provide four parallel testing modes to achieve fine-grained testing efficiency optimization: 1) mode-1: N cpu cores are used to distribute the N testing problem instances, on each core, line2-line3 are executed as two-layer loop; 2) mode-2: R cpu cores are used to distribute the R independent runs, while on each core, line1-line2 are executed as two-layer loop; 3) mode-3: NxB cpu cores are used to distribute the N problem instances and B baselines, while on each core, line3 are executed as one-layer loop; 4) mode-4: NxBxR cpu cores are used to distribute all evaluation tasks, hence there is no loop anymore. The hardware requirement from mode-1 to mode-4 is incremental.

By decomposing parallelism into two orthogonal dimensions as above: (a) distributed solving across *problem instances*, and (b) parallel execution of *independent test runs*, we provide sufficient flexibility for users to accelerate their programs based on their specific hardware conditions.

### 3.3 Novel Evaluation Metrics

**Metadata.** MetaBox-v2 inherits the automatic train-test-log workflow of original MetaBox. However, the original MetaBox does not provide interfaces for users to operate on the process data observed from both the training and testing. Instead, it provides users post-processed data such as comparison tables and optimization progress figures. As an emerging topic, it is still an open question how to measure different MetaBBO approaches with fairness and objectivity. To this end, we open a data acquirement interface $\mathrm{get\_metadata}()$ for users who would like to custom their own metrics in analysis. For example, consider evaluating a pre-trained MetaBBO approach $\mathcal{A}$ on a testsuite $\mathbb{D}$ containing $N$ problem instances $\{p_1, ..., p_N\}$. For each problem instance $p_i$, we execute $K$ independent runs. Then the metadata $md_i$ for $p_i$ is structured as a json object:

$$\{\texttt{"problem\_id"}:p_i, \texttt{ "data"}:\{ \texttt{"run\_1"}:\{\texttt{"X"}:\mathrm{List},\texttt{"Y"}:\mathrm{List},\texttt{"T"}:5.23\}, \\ \cdots, \\ \texttt{"run\_K"}:\{\texttt{"X"}:\mathrm{List},\texttt{"Y"}:\mathrm{List},\texttt{"T"}:4.96\} \}\} \tag{1}$$

where `"X"` is a list of each generation's solution population, `"Y"` is a list of the objective values, `"T"` is the wall-clock time consumed for optimizing $p_i$. Then the overall metadata $md(\mathcal{A}, \mathbb{D})$ is aggregated from each $p_i$: $\{\texttt{"problem\_type"}:\mathrm{SOO}, \texttt{ "all\_data"}:[md_1, ..., md_N]\}$, where `"problem_type"` is the optimization types of $\mathbb{D}$. Notably, $md(\mathcal{A}, \mathbb{D})$ provide significant convenience for computing various metrics. For example, a normalized performance metric widely used in existing MetaBBO approaches [13, 15, 31, 70] can be easily computed as $\mathrm{Perf}(\mathcal{A}, \mathbb{D}) = \frac{1}{N \times K}[\sum_{i=1}^{N} \sum_{j=1}^{K} \frac{Y_{i,j}^* - p_i^*}{Y_{i,j}^0 - p_i^*}]$, where $Y_{i,j}^0$ and $Y_{i,j}^*$ are the initial and final objective values, $p_i^*$ is the optimal. We next showcase two customized metrics based on the metadata.

**Learning Efficiency Indicator.** We provide a novel built-in metric in MetaBox-v2: *learning efficiency*, to measure how efficiently a MetaBBO approach learns an effective meta-level policy. Specifically, during the training of a given algorithm $\mathcal{A}$, we save a series of its model snapshots $\{\mathcal{A}^{(g)}\}_{g=0}^{G}$ where $G$ is the number of training epochs. Evaluating all snapshots on the testsuite $\mathbb{D}$, we obtain corresponding metadata: $\{md(\mathcal{A}^{(g)}, \mathbb{D})\}_{g=0}^{G}$. Suppose training $\mathcal{A}^{(g)}$ consumes $T^{(g)}$ hours, then the *learning efficiency* of $\mathcal{A}^{(g)}$ is computed as: $\frac{\mathrm{Perf}(\mathcal{A}^{(g)}, \mathbb{D})}{T^{(g)}}$. This metric could fairly reflect the training efficiency of $\mathcal{A}$ at different time slots $g$.

**Anti-NFL Indicator.** Recall that the core motivation of MetaBBO is to learn generalizable policy towards unseen problems. This is somewhat against the well-known *no-free-lunch* (NFL) theorem [5]. We hence propose a novel indicator named as $\mathrm{Anti\text{-}NFL}$, which could reflect the performance variance of a given algorithm $\mathcal{A}$ on unseen testsuites apart from the one it was trained. Suppose $\mathcal{A}$ is trained on $\mathbb{D}_{\mathrm{train}}$, and tested on other $B$ testsuites: $\{\mathbb{D}_{\mathrm{test}}^{(b)}\}_{b=1}^{B}$. After obtaining all of the metadata by testing the final model $\mathcal{A}^{(G)}$ on these testsuites, the $\mathrm{Anti\text{-}NFL}$ is computed as:

$$\mathrm{Anti\text{-}NFL} = \exp\left( \frac{1}{B} \sum_{b=1}^{B} \frac{\mathrm{Perf}\left(\mathcal{A}^{(G)}, \mathbb{D}_{\mathrm{test}}^{(b)}\right) - \mathrm{Perf}\left(\mathcal{A}^{(G)}, \mathbb{D}_{\mathrm{train}}\right)}{\mathrm{Perf}\left(\mathcal{A}^{(G)}, \mathbb{D}_{\mathrm{train}}\right)} \right) \tag{2}$$

A larger $\mathrm{Anti\text{-}NFL}$ indicator indicates that $\mathcal{A}$ performs robustly under problem-shifts, and vice versa.

## 4 Benchmarking Study

We present a comprehensive case study on up-to-date MetaBBO approaches through MetaBox-v2, addressing four critical research questions: **RQ1:** Does MetaBox-v2 significantly enhance train/test efficiency compared to the conventional version? **RQ2:** How effectively do up-to-date MetaBBO methods generalize under standardized training protocols and diverse test scenarios? **RQ3:** How to objectively compare the dominance relationship of learning efficiency and generalization ability of baselines? **RQ4:** How does MetaBBO perform under extreme problem shift in practice?

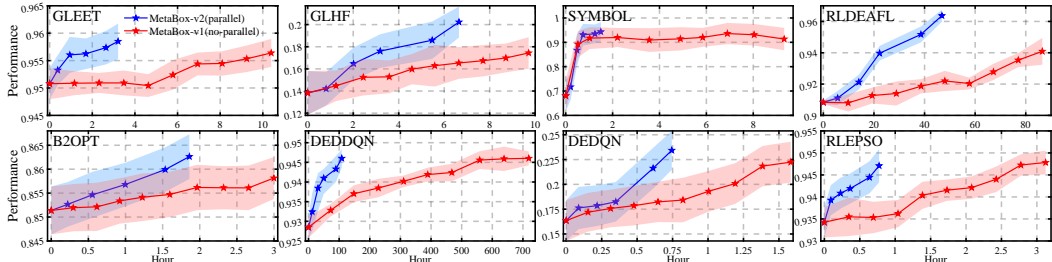

Figure 4: Training improvement curves of baselines on Metabox-v2 and MetaBox.

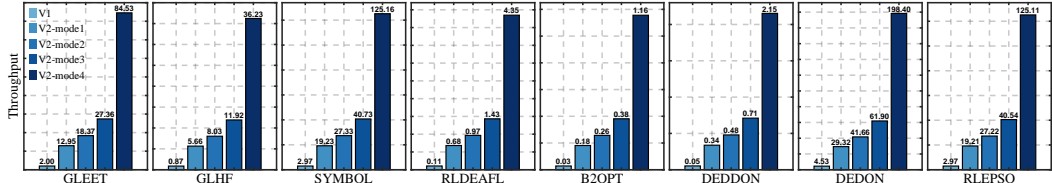

Figure 5: Testing efficiency comparison of MetaBox-v2 (4 parallel modes) and MetaBox.

## 4.1 Experimental Setup

**Baselines.** We select 20 baselines from the library of MetaBox-v2 for case study, including 5 traditional BBO optimizers: PSO [2], DE [3], SHADE [71], JDE21 [72], MadDE [73], and 15 up-to-date MetaBBO baselines from all of four learning paradigms: RNNOPT [11], DEDDQN [27], DEDQN [29], LDE [28], RLPSO [74], RLEPSO [30], NRLPSO [52], LES [15], GLEET [31], GLHF [13], RLDAS [32], SYMBOL [33], OPRO [17], B2OPT [12], RLDEAFL [36]. The settings follows their original papers.

**Testsuites.** First, we employ the *bbob-10D* testsuite and split its 24 problem instances into 8 training instances and 16 testing instances, with the latter serving as in-distribution evaluation. Then, the out-of-distribution evaluation involves four other testsuites: *bbob-noisy-30D*, *protein*, *uav*, and *hpo-b*. To ensure the fairness of training, all algorithms undergo 1600 episodes for each training instance. The testing phases employ 51 independent runs with seed-controlled reproducibility. All experiments are conducted with 2 AMD EPYC 7H12 CPU, a RTX 3080 GPU and 512GB RAM.

## 4.2 Platform's Acceleration Performance (RQ1)

We use vectorized environment with *batch_size* as 16 to accelerate the training of all involved MetaBBO baselines. Due to the space limitation, we selectively illustrate in Figure 4 the performance improvement curves of 8 baselines, where y-axis denotes normalized performance on testing set of *bbob-10D*. Compared to original MetaBox, MetaBox-v2 consistently accelerates MetaBBO baselines by at most 10x. We can observe that the concrete acceleration may varies on different baselines, this is because the differences of the internal logic and communication cost among the baselines. An important note is that the irregular record points in Figure 4 is due to the unstable multi-processing. For parallel cases, we draw 5 points (every 20 training epochs), which might present irregular patterns since the x-axis denotes the training time.

We also illustrate the acceleration performance of MetaBox-v2 compared to original MetaBox in Figure 5 in terms of testing efficiency, where y-axis denotes the throughput of evaluation process measured by the number of instance test runs per second. We compare the throughput of the 4 Ray modes in MetaBox-v2 to original MetaBox, and the results show that even the simplest distribution *Mode-1* could significantly accelerate the testing workflow. If users have advanced hardware, the distribution *Mode-4* could introduce no less than 40x acceleration.

## 4.3 Generalization Performance Comparisons among Baselines (RQ2)

**In-distribution Test.** Table 3 shows the average results and error bars on the testing set of *bbob-10D* across 51 independent runs. We additionally summarize the average ranks among the baselines at the bottom of the table. The in-distribution test aims at validating the basic learning effectiveness of

Table 3: In-distribution optimization performances of baselines over *bbob-10D*, with gray box labeling the best. Due to the space limitation, results for 8 of 16 problem instances in *bbob-10D*'s testing set are presented here while the complete results can be accessed at this online page.

| | Sharp_Ridge | Different_Powers | Schaffers_HC | Composite_GR | Schwefel | Gallagher_21 | Katsuura | Lunacek_BR |
|---|---|---|---|---|---|---|---|---|
| PSO(1995) [2] | 1.905E+02 ± 2.156E+01 | 6.802E-01 ± 1.760E-01 | 5.600E+00 ± 1.368E+00 | 3.290E+00 ± 5.796E-01 | 2.560E+00 ± 3.067E-01 | 6.803E+00 ± 6.472E+00 | 1.272E+00 ± 2.933E-01 | 6.139E+01 ± 5.747E+00 |
| DE(1997) [3] | 8.588E-01 ± 1.054E+00 | 8.180E-04 ± 2.537E-04 | 9.454E-02 ± 6.483E-02 | 2.577E+00 ± 4.860E-01 | 9.156E-01 ± 3.039E-01 | 3.393E+00 ± 4.999E+00 | 1.467E+00 ± 2.734E-01 | 4.210E+01 ± 3.043E+00 |
| SHADE(2013) [71] | 1.442E+00 ± 4.321E-01 | 2.721E-04 ± 4.192E-05 | 2.649E-01 ± 6.818E-02 | 2.238E+00 ± 3.476E-01 | 1.338E+00 ± 1.957E-01 | 1.155E+00 ± 9.320E-01 | 1.553E+00 ± 3.454E-01 | 4.248E+01 ± 4.209E+00 |
| JDE21(2021) [72] | 3.476E+00 ± 6.350E+00 | 4.398E-04 ± 3.807E-04 | 4.496E-01 ± 3.700E-01 | 2.542E+00 ± 6.355E-01 | 5.777E-01 ± 2.246E-01 | 1.604E+00 ± 1.641E+00 | 1.416E+00 ± 3.359E-01 | 4.059E+01 ± 7.940E+00 |
| MADDE(2021) [73] | 1.736E+00 ± 3.300E-01 | 5.830E-04 ± 2.318E-04 | 9.538E-01 ± 2.897E-01 | 1.077E+00 ± 3.709E-01 | 8.049E-01 ± 1.997E-01 | 5.458E-01 ± 7.264E-01 | 1.350E+00 ± 2.395E-01 | 4.308E+01 ± 4.974E+00 |
| RNNOPT(2017) [11] | 1.822E+03 ±0.000E+00 | 2.297E+01 ±0.000E+00 | 4.645E+01 ±0.000E+00 | 3.609E+00 ±0.000E+00 | 9.297E+03 ±1.819E-12 | 8.431E+01 ±0.000E+00 | 2.186E+00 ±0.000E+00 | 1.142E+02 ±0.000E+00 |
| DEDDQN(2019) [27] | **1.841E-03** ±**1.841E-03** | **4.224E-09** ±**4.069E-09** | **1.080E-02** ±**7.097E-03** | 2.480E+00 ±5.250E-01 | 1.720E+00 ±4.164E-01 | 1.574E+00 ±9.236E-01 | 1.344E+00 ±2.839E-01 | 4.039E+01 ±4.264E+00 |
| DEDQN(2021) [29] | 9.538E+02 ±1.548E+02 | 1.115E+01 ±2.837E+00 | 2.709E+01 ±5.790E+00 | 1.268E+01 ±2.131E+00 | 4.880E+03 ±3.385E+03 | 5.711E+01 ±1.366E+01 | 3.286E+00 ±6.136E-01 | 1.591E+02 ±2.132E+01 |
| LDE(2021) [28] | 5.955E-01 ±5.103E-01 | 5.159E-05 ±3.700E-05 | 2.156E-01 ±1.238E-01 | 2.024E+00 ±1.812E-01 | 1.071E+00 ±1.603E-01 | **4.292E-01** ±**7.059E-01** | 1.306E+00 ±2.245E-01 | 3.616E+01 ±3.494E+00 |
| RLPSO(2021) [74] | 2.769E+02 ±7.000E+01 | 1.481E+00 ±9.514E-01 | 1.429E+01 ±2.968E+00 | 3.629E+00 ±1.115E+00 | 2.722E+00 ±2.998E-01 | 1.597E+01 ±1.719E+01 | 2.225E+00 ±3.550E-01 | 6.525E+01 ±7.460E+00 |
| RLEPSO(2022) [30] | 6.388E+00 ±6.093E+00 | 2.554E-04 ±1.396E-04 | 1.687E+00 ±7.471E-01 | 1.387E+00 ±4.516E-01 | 1.261E+00 ±2.497E-01 | 7.703E+00 ±1.223E+01 | 1.017E+00 ±2.993E-01 | **2.413E+01** ±**7.015E+00** |
| NRLPSO(2023) [52] | 1.968E+02 ±8.105E+01 | 6.449E-01 ±3.607E-01 | 5.710E+00 ±2.194E+00 | 3.367E+00 ±1.081E+00 | 2.631E+00 ±4.837E-01 | 7.478E+00 ±5.155E+00 | 1.599E+00 ±4.433E-01 | 7.007E+01 ±1.466E+01 |
| LES(2023) [15] | 1.099E+03 ±1.516E+02 | 1.273E+01 ±2.222E+00 | 3.812E+01 ±6.523E+00 | 1.215E+01 ±2.095E+00 | 8.044E+03 ±4.741E+03 | 5.777E+01 ±2.074E+01 | 4.099E+00 ±9.875E-01 | 1.793E+02 ±2.481E+01 |
| GLEET(2024) [31] | 4.464E+00 ±7.370E+00 | 1.130E-04 ±8.072E-05 | 2.137E+00 ±1.618E+00 | **8.624E-01** ±**3.202E-01** | 1.481E+00 ±1.765E-01 | 8.632E+00 ±1.209E+01 | **4.839E-01** ±**2.181E-01** | 2.717E+01 ±8.473E+00 |
| GLHF(2024) [13] | 9.652E+02 ±1.286E+02 | 1.074E+01 ±1.796E+00 | 3.163E+01 ±5.541E+00 | 1.027E+01 ±1.857E+00 | 6.827E+03 ±4.036E+03 | 4.923E+01 ±1.678E+01 | 3.527E+00 ±9.244E-01 | 1.582E+02 ±2.103E+01 |
| RLDAS(2024) [32] | 1.627E+00 ±1.073E+00 | 3.740E-04 ±2.542E-04 | 9.798E-01 ±5.450E-01 | 1.650E+00 ±4.859E-01 | **5.505E-01** ±**3.074E-01** | 4.698E-01 ±7.563E-01 | 1.296E+00 ±2.623E-01 | 3.630E+01 ±1.035E+01 |
| SYMBOL(2024) [33] | 1.344E+01 ±9.453E+00 | 5.332E-03 ±2.537E-03 | 4.256E+00 ±2.238E+00 | 1.383E+00 ±4.880E-01 | 1.732E+00 ±2.436E-01 | 5.611E+00 ±4.981E+00 | 6.371E-01 ±3.070E-01 | 3.188E+01 ±1.164E+01 |
| OPRO(2024) [17] | 2.003E+03 ± 1.562E+02 | 3.007E+01 ± 3.326E+00 | 5.099E+01 ± 9.258E+00 | 1.434E+01 ± 6.317E+00 | 9.299E+03 ± 4.804E+03 | 9.031E+01 ± 2.184E+01 | 6.113E+00 ± 2.771E+00 | 1.812E+02 ± 2.562E+01 |
| B2OPT(2025) [12] | 2.581E+02 ±3.724E+00 | 2.510E+00 ±3.641E+01 | 6.057E+00 ±1.696E+00 | 8.728E-01 ±2.494E-01 | 2.543E+00 ±1.547E-01 | 1.130E+01 ±7.585E+00 | 1.575E+00 ±2.701E-01 | 5.814E+01 ±6.917E+00 |
| RLDEAFL(2025) [36] | 1.136E+01 ±1.345E+01 | 1.487E-04 ±9.165E-05 | 4.166E+00 ±2.214E+00 | 2.535E+00 ±1.036E+00 | 1.397E+00 ±3.326E-01 | 5.452E+00 ±6.106E+00 | 1.199E+00 ±6.034E-01 | 3.231E+01 ±7.111E+00 |
| Rank | 1:LDE, 2:DEDDQN, 3:RLDAS, 4:SHADE, 5:MADDE, 6:GLEET, 7:RLEPSO, 8:RLDEAFL, 9:JDE21,10:DE, 11:SYMBOL, 12:PSO, 13:B2OPT, 14:NRLPSO, 15:RLPSO, 16:GLHF, 16:DEDQN, 18:RNNOPT, 19:LES, 20:OPRO | | | | | | | |

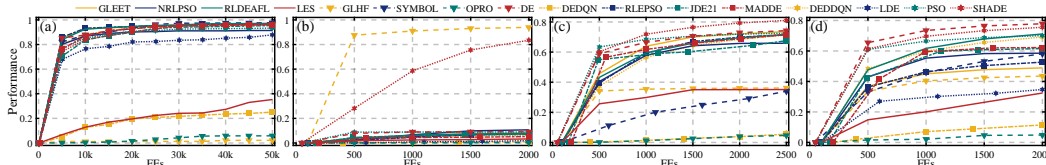

Figure 6: Out-of-distribution generalization performance of baselines on: (a) *bbob-noisy-30D*; (b) *protein*; (c) *uav*; and (d) *hpob*.

MetaBBO since the synthetic problem instances within *bbob-10D* show certain similarity in landscape features [75]. Several key observations can be obtained: 1) Overall, on 14 of all 16 testing instances, MetaBBO baselines attain the best optimization results and advance traditional BBO baselines by orders of magnitudes. 2) So far, the MetaBBO-RL baselines generally outperform MetaBBO-SL, MetaBBO-NE and MetaBBO-ICL baselines. 3) We notice that a 2019 method DEDDQN [27] outperforms other baselines in 6 of 18 testing instances and ranks the first place on average.

**Out-of-distribution Test.** Figure 6 presents comparative evaluation results across MetaBBO and traditional BBO baselines using four out-of-distribution testsuites (*bbob-noisy-30D*, *protein*, *uav*, and *hpo-b*), with the y-axis representing instance-normalized performance averages. Combining the results of in-distribution test, several valuable insights are obtained: 1) Generally speaking, traditional BBO algorithms such as DE, SHADE demonstrate empirical robustness across testsuites, contrasting with MetaBBO baselines that appear to overestimate their generalization potential; 2) The out-of-distribution performance of DEDDQN on *protein* is reversed from its good performance in the in-distribution test, which might indicates that the overfitting issue should be addressed for future MetaBBO researches. 3) Almost all of MetaBBO baselines show certain level of performance oscillation in diverse testsuites, which further underscores that how to define and measure similarity across diverse BBO problems is crucial to ensure MetaBBO's generalization.

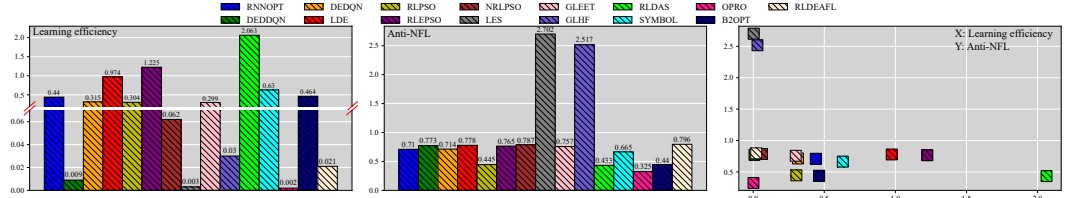

Figure 7: Left: Learning efficiency comparison of MetaBBO baselines, larger is better. Middle: Anti-NFL indicator of MetaBBO baselines, larger is better. Right: Domination relationship among MetaBBO baselines considering learning efficiency and Anti-NFL indicator.

## 4.4 Other In-depth Analysis

Although there is continuous discussion on the fairness and objectivity of BBO benchmarks [76–78], it is still an open challenge for optimization community to agree on a certain golden standard, let along for the MetaBBO field. We hence provide the following discussions on the objective profiling of efficiency and generalization of MetaBBO, and the impact analysis of extreme problem shift.

**Learning Efficiency (RQ3).** Following the computation detail introduced in Section 3.3, we compute the *learning efficiency* indicators of all baselines on all testsuites. Their average efficiency values is shown in the left of Figure 7. Combining the results with the average ranks of baselines in Table 3, we could conclude that RLDAS [32] is a remarkable MetaBBO baseline since it uses less computational resource to achieve better optimization performance. In contrast, DEDDQN [27] achieves the best performance while consuming hundreds of hours for training, which might not be favorable when the computational resource is limited. It is also worthy to note that MetaBBO-NE approach such as LES [15] and MetaBBO-ICL approach such as OPRO [17] hold the lowest efficiency due to the nested evolutionary optimization and the expensive LLM calling, respectively.

**Anti-NFL Performance (RQ3).** The Anti-NFL indicator computed in Eq. 2 reflects the robustness of a MetaBBO approach when being generalized to diverse BBO problems. The middle of Figure 7 reports the Anti-NFL indicators of MetaBBO baselines on all testsuites. The conclusions could be obtained here seems to be different with the aforementioned performance metrics. Two MetaBBO baselines GLHF [13] and LES [15] have much higher Anti-NFL values than others, while they hold relatively low absolute performance (Table 3). This point deserves in-depth analysis in future works. While RLDAS has favorable performance and efficiency, its Anti-NFL is among the lowest due to its meta-level policy's architecture, which is not generalizable across different problem dimensions.

Combining the results of *learning efficiency* and Anti-NFL indicator, as illustrated in the right of Figure 3, we analyze the domination relationship of MetaBBO baselines. It can be observed that, so far, no baseline participating in this case study dominates all the others. This reflects that there is certain design tradeoff in existing MetaBBO between the efficiency and effectiveness.

**Extreme Problem Shift Analysis (RQ4).** We evaluate MetaBBO's adaptability through an extreme domain shift scenario: algorithms trained on low-dimensional synthetic problems (*bbob-10D*) are directly deployed on high-dimensional neuroevolution tasks (*ne*), a robotic control benchmark integrated from EvoX [51] where optimizers must tune neural networks (thousands of parameters) to maximize robotic returns. The results from three Ant tasks (Ant-3, Ant-4, Ant-5 with 4328, 5384, 6440 parameters) are reported in Figure 8, which reveal that: 1) Certain MetaBBO methods (e.g., GLEET [31] with Transformer meta-policy) achieve performance parity or superiority over advanced BBO baselines (CMAES [4] and GLPSO [79]) despite the exclusive training on simple problems. 2) MetaBBO performance correlates with policy architecture complexity—Transformer-based GLEET maintains robustness across scaling dimensions, while MLP (RLEPSO [30]) and LSTM (LDE [28]) agents degrade sharply in higher-dimensional tasks.

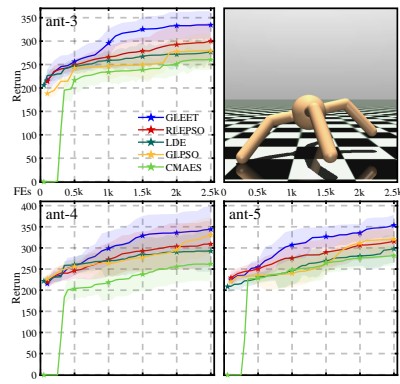

Figure 8: Comparison on neuroevolution task by return curves.

# 5 Discussion

**Takeaways.** This work proposes MetaBox-v2 as a milestone upgrade for its predecessor, introducing several key architecture adjustments that establish a comprehensive benchmark platform for the MetaBBO research. The fundemental advancements not only enable unified development and evaluation for various MetaBBO paradigms and diverse optimization problem types; but also support streamlined parallel acceleration for training/evaluation by 10x-40x. With the expanded baseline library ($8 \rightarrow 23$) and testsuite library ($3 \rightarrow 18$, $1900+$ problem instances), we conduct a rigorous case study, which discloses insights including but not limited to: 1) Current literature overestimates MetaBBO capabilities through narrow evaluation practices, with MetaBox-v2 exposing substantial performance gaps in cross-domain settings. 2) Out-of-distribution generalization demands special attention, since our analysis reveals that overfitting persists across baseline algorithms. 3) Effective MetaBBO assessment requires multidimensional analysis (optimization efficacy, learning efficiency, generalization robustness) beyond traditional convergence metrics.

**Future Work.** We outline the following directions to continuously improve MetaBox-v2: 1) Maintain cutting-edge benchmarks through continuous integration of emerging algorithms and problem suites, with an open-source ecosystem welcoming community contributions; 2) Optimize the parallel computing framework to achieve higher resource utilization; 3) Lower adoption barriers through enhanced tutorials and beginner-friendly interfaces. We aim to establish MetaBox-v2 as both a powerful research platform and an accessible educational resource for the MetaBBO field.

## Acknowledgments and Disclosure of Funding

This work was supported in part by National Natural Science Foundation of China (Grant No. 62276100), in part by Guangzhou Science and Technology Elite Talent Leading Program for Basic and Applied Basic Research (Grant No. SL2024A04J01361), in part by the Guangdong Provincial Natural Science Foundation for Outstanding Youth Team Project (Grant No. 2024B1515040010), and in part by the Fundamental Research Funds for the Central Universities (Grant No. 2025ZYGXZR027).

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
