# OpenReview forum: "MetaBox-v2: A Unified Benchmark Platform for Meta-Black-Box Optimization"
_NeurIPS.cc/2025/Datasets_and_Benchmarks_Track — NeurIPS 2025 Datasets and Benchmarks Track poster_

### Official Review · Reviewer_xaLo · 2025-06-27

**Rating:** 5
**Confidence:** 4

**Summary:**

This paper presents MetaBox-v2, a significant upgrade to the original MetaBox platform for Meta-Black-Box Optimization (MetaBBO). The authors propose four major improvements: 1) unified support for all MetaBBO paradigms (RL, SL, NE, ICL), 2) efficient parallelization achieving 10-40x speedup, 3) comprehensive benchmarks with 18 testsuites and 1900+ problem instances, and 4) extensible interfaces for custom analysis. A comprehensive benchmarking study evaluates 20 baseline methods across diverse optimization scenarios.

**Dataset Code Accessibility:**

Yes

**Ethical Considerations:**

No, there are no or only very minor ethics concerns

**Final Justification:**

The authors have addressed my main concerns in their rebuttal, particularly by clarifying the rationale for the distribution of baselines, detailing support for MOO metrics, and committing to additional experiments and manuscript clarifications. While some limitations remain in the breadth of evaluation for non-SOO scenarios, the authors provided reasonable justifications and outlined plans for future improvements. Given these responses and the overall technical merit of the work, I believe the paper meets the NeurIPS acceptance bar and update my positive score.

**Limitations Weaknesses:**

### **Major concerns**

- While claiming comprehensive support for all paradigms, the platform shows clear bias: MetaBBO-RL methods (10) are well-developed, but support for MetaBBO-SL (2), MetaBBO-NE (2), and MetaBBO-ICL (1) needs to be substantially developed. More critically, despite claiming to support diverse optimization scenarios (SOO, MOO, MMO, MTO, LSGO) with 1900+ problem instances, the experimental analysis exclusively focuses on SOO problems. This omission makes it impossible to assess whether the proposed platform improvements actually benefit the other optimization scenarios, undermining the paper's claims of comprehensive coverage.
- For multi-objective optimization (MOO), mature libraries like pymoo already provide comprehensive benchmarks and algorithms. The paper's MOO problems appear to be a subset of what pymoo offers, yet there's no discussion of why they didn't build interfaces to pymoo for secondary development rather than reimplementing. More importantly, MOO requires fundamentally different evaluation metrics than SOO (e.g., hypervolume, IGD, spread) which are not addressed in the proposed AEI metric or experimental analysis. This oversight raises questions about the platform's actual readiness for non-SOO scenarios and suggests the authors should either properly support MOO-specific evaluation or clearly scope their contribution to SOO enhancement.


### **Other concerns**
- The paper introduces Perf() function in the learning efficiency and Anti-NFL indicators without proper definition. This creates a fundamental interpretation problem: Table 3 shows optimization results where smaller values are better (minimization), while Figure 6 shows performance curves that increase over time (suggesting maximization).
- Figure 4 shows learning efficiency points with irregular intervals but doesn't explain the sampling strategy.
- The Learning Efficiency indicator (performance/time) lacks practical meaning.

**Strengths Contributions:**

Compared with the first version, MetaBox-v2 has the following strengths.
- Broader Paradigm Support: The extension from RL-only to all four MetaBBO paradigms is a genuine improvement, though the implementation depth varies significantly across paradigms.
- Expanded Problem Coverage: The increase from 334 to 1900+ problem instances across diverse optimization scenarios (SOO, MOO, MMO, MTO, LSGO) represents substantial progress.
- Parallelization: The introduction of vectorized environments and Ray-based parallelization addresses a critical bottleneck absent in v1.

---

> ### Author Rebuttal · Authors · 2025-07-30
>
> We want to express our deepest gratitude for your positive feedback and recognition of our efforts in MetaBox's expansion and parallelization. We provide following point-to-point responses to address your constructive comments.
>
> ### [W1: Unbalanced Distribution of Baseline&Testsuites]
>
> We appreciate your constructive insights. We would like to first clarify that the primary efforts we have made in MetaBox-v2 is to bridge diverse MetaBBO paradigms, providing easy enough and universal experimental protocols for either practitioners or newcomers to learn, use or even develop diverse MetaBBO approaches. To this end, we have update the MetaBox-v1 by a) universal training and testing interfaces, b) coding examples of different MetaBBO types (at least one for each type), and c) common optimization problem types (at least one for each optimization domain). We hope this primary contribution could be recognized. Let us further explain several aspects for your concerns:
>
> - The unbalanced distribution of different MetaBBO paradigms in MetaBox-v2 actually objectively reflect the literature number on each category. We sincerely invite the reviewer to take a brief glance at two latest surveys [1][2] on MetaBBO. MetaBBO-RL on SOO, according these two surveys, is a core research avenue in MetaBBO, while MetaBBO-NE and MetaBBO-SL works are still limited. For these two categories, we have implemented the most known baselines. For MetaBBO-ICL, it is an emerging avenue with growing interests indeed. However, most of works in this line such as FunSearch[3], EoH[4] and Reevo[5] are tailored for combinatorial problems, resulting in inherent gap applying them on BBO problem, which are normally continuous single/multi-objective ones.
> - For all baselines and their target optimization problem types we have inlucded, the efficiency and performance reproducibility tests are all conducted. The efficiency improvement of MetaBBOs for MOO, MTO, LSGO etc. resembles the results of SOO we presented in the paper. We will include these results in the appendix of the revised paper. The primary choice of SOO as a case study is the sufficient baselines and testsuites facilitate systematic and in-depth analysis.
> - As the capacity of MetaBox-v2 extended by our continuous update and maintainence, more systematic study on diverse domains is deemed as a promising future work.
>
> [1] Yang X, Wang R, Li K, et al. Meta-Black-Box optimization for evolutionary algorithms: Review and perspective[J]. Swarm and Evolutionary Computation, 2025, 93: 101838.
>
> [2] Ma Z, Guo H, Gong Y J, et al. Toward automated algorithm design: A survey and practical guide to meta-black-box-optimization[J]. IEEE Transactions on Evolutionary Computation, 2025.
>
> [3] Romera-Paredes B, Barekatain M, Novikov A, et al. Mathematical discoveries from program search with large language models[J]. Nature, 2024, 625(7995): 468-475.
>
> [4] Liu F, Tong X, Yuan M, et al. Evolution of heuristics: Towards efficient automatic algorithm design using large language model[J]. arXiv preprint arXiv:2401.02051, 2024.
>
> [5] Ye H, Wang J, Cao Z, et al. Reevo: Large language models as hyper-heuristics with reflective evolution[J]. Advances in neural information processing systems, 2024, 37: 43571-43608.
>
> ### [W2: MOO Support]
>
> - There are testsuites in MetaBox-v2 build upon external interfaces (such as neuroevolution tasks in EvoX). There are also testsuites that we build by our own such the MOO testsuites you mentioned. The reason behind is that by showing users such two cases, they flexibly develop their own testing problem either by self-coding or calling interfaces. As we disclosured in the end of this paper, given the MetaBox-v2’s universal interfaces, extendability and compatibility toward external resources, we will continuously pay attention to cutting-edge methods and testsuites to maintain an active ecosystem for MetaBBO. Strong testsuites like pymoo is clearly located at our agenda!
> - Let us explain for you how MetaBox-v2 support MetaBBO for MOO methods. First, in MetaBox-v2, we specify IGD as the performance indicator of multi-objective problems by default, which serve as the role of objective value in SOO. By doing so, post-processing indicators such as AEI can be automatically calculated accordingly. Furthermore, we have a specific interface for MOO users to indicate their preference on the indicators HV, IGD etc. In the metadata of MOO testing, these MOO indicators are all recorded for users to use. We will add our preliminary testing results on MOO baseline MADAC into appendix to show MOO support and update our documentation to notify users about this.
>
> ### [W3: Perf() Metric]
>
> - We apologize for the missing definition of $Perf()$. Given a BBO optimizer or MetaBBO baseline $\mathcal{A}$ and a test problem set $D$ (w.o.l.g., all instances are minimization problems),  $Perf(\mathcal{A},D)$ measures normalized optimization performance of $\mathcal{A}$ averaged over all instances in $D$:
>
>     $$
>     Perf\left(\mathcal{A}, \mathbb{D}\right) = 1 - \frac{1}{|\mathbb{D}|}\frac{1}{51}\sum\_{p\in\mathbb{D}}\sum\_{i=1}^{51}\frac{y^\*\_{p,i} - y^*\_p}{y^{(0)}\_{p,i} - y^\*\_p}
>     $$
>
>
> Here we run $\mathcal{A}$ to optimize each instance in $D$ for independent 51 runs. $y^\*\_{p,i}$ denotes the  best objective value found in $i$-th run of instance $p$. $y^{(0)}\_{p,i}$ denotes the best objective value within the initialized population in $i$-th run of instance $p$. $y^*\_p$ denotes the optimal objective of instance $p$ itself. We will refine this definition in the revised paper.
>
> - Table 3 shows the cost of each algorithm, which is very important for many BBO researchers. The visualization chart in Figure 5 makes it easier for people in other fields to clearly understand the differences between the algorithms. We hope that these two forms of expression can better meet the needs of various groups of people. Of course, this is also caused by the lack of a clear definition of Perf(). After this work is accepted, we will make a clearer statement in the title of Table 3 and Figure 5, and explicitly use cost instead of performance to avoid confusion.
>
> ### [W4: Irregular Points in Figure 4]
>
> The sample strategy is based on training epochs. For all baselines, we train them in MetaBox-v2 for 100 epochs. For parallel cases (blue lines), we sample every 20 epochs. The irregular cases that happen at parallel training cases (blue lines) come from the unstable multi-processing of our computers for experiments. During the experiments, there are several training/testing tasks running simutaneously, which leads to relatively significant impact on computation-intensive multi-processing in parallel cases. We have now re-conducted the same experiments and will revise Figure 4 once our paper was accepted. We appreciate your careful reading to help us improve our paper’s quality.
>
> ### [W5: Learning Efficiency]
>
> We would like to argue that the learning efficiency metric could give an objective and intuitive perspective for researchers to compare their MetaBBO approach with the other baselines. Given that all baselines are trained by the same hardware settings, this metric more or less reflects how much computational resources consumed by a MetaBBO approach to attain performance improvement. That is, to say, as long as all baselines are compared by the same computational settings,   performance/time can naturally characterize the performance improvement of each baseline per unit computational resources.
>
> We hope all of the responses above could address your concerns. Looking forward to youe timely feedback. Any further questions are welcomed!

---

> > ### Comment · Reviewer_xaLo · 2025-08-04
> >
> > My concerns have been addressed by the authors’ thorough response and their commitment to incorporating additional experiments and clarifications in the revised manuscript. I believe this paper now meets the acceptance bar of NeurIPS, so I maintain my positive score. Assuming the promised revisions are made, the work should make a valuable contribution to the MetaBBO community and adequately resolves my earlier reservations regarding scope and clarity.

---

### Official Review · Reviewer_oUsU · 2025-07-01

**Rating:** 5
**Confidence:** 3

**Summary:**

MetaBox-v2 is an enhanced open-source platform for Meta-Black-Box Optimization (MetaBBO), advancing automated algorithm design through meta-learning. Upgrading MetaBox (2023), it introduces: (1) a unified framework supporting RL, SL, neuroevolution, and in-context learning, expanding baselines from 8 to 23; (2) parallelization schemes, speeding up training/testing by 10-40x; (3) 18 diverse tasks (1900+ instances) spanning single/multi-objective, multi-modal, and multi-task optimization; and (4) extensible interfaces for analysis and tool integration. A case study of 20 baselines shows strong in-distribution performance but out-of-distribution generalization challenges. Available on GitHub with comprehensive documentation, MetaBox-v2 is a robust tool for MetaBBO research and education.

**Dataset Code Accessibility:**

Yes

**Ethical Considerations:**

No, there are no or only very minor ethics concerns

**Limitations Weaknesses:**

1. Given the limited inclusion of neuroevolution (MetaBBO-NE) and LLM-based methods in MetaBox-v2 and the rising popularity of AlphaEvolve, expanding the platform to incorporate more advanced approaches could enhance its benchmarking capabilities and align with current trends in MetaBBO research.
2. Viewing the GitHub repository, no testing workflow was found to ensure code functionality after updates. Such testing is crucial for maintaining the reliability of a large number of benchmarks.

**Strengths Contributions:**

1. Algorithmic Flexibility and Integration: MetaBox-v2 provides a unified framework supporting four MetaBBO paradigms—reinforcement learning, supervised learning, neuroevolution, and in-context learning—enabling seamless algorithm development and expanding the baseline library from 8 to 23 algorithms.
2. Performance and Scalability: By implementing vectorized optimization and instance-level distributed evaluation, MetaBox-v2 achieves a 10-40x reduction in training and testing time, enhancing computational efficiency for large-scale experiments.
3. Comprehensive Benchmarking and Extensibility: The platform features a diverse test suite of 18 tasks with over 1900 instances, covering single-objective, multi-objective, multi-modal, and multi-task optimization, alongside extensible interfaces for custom analysis and integration with external tools.
Together, these advancements, supported by rigorous case studies and open-source documentation, establish MetaBox-v2 as a powerful and accessible platform for advancing MetaBBO research.

---

> ### Author Rebuttal · Authors · 2025-07-30
>
> Dear reviewer, thank you for your careful reading and professional suggestions！We are honored that you have recognized the flexibility and scalability of our work, which are important for the development of the MetaBBO. Next we will respond to your concerns in detail.
>
> ### [W1: Inclusion of Diverse MetaBBOs]
>
> We appreciate your insights on our platform. We would like to first clarify that the primary efforts we have made in MetaBox-v2 is to bridge diverse MetaBBO paradigms, providing easy enough and universal experimental protocols for either practitioners or newcomers to learn, use or even develop diverse MetaBBO approaches. To this end, we have update the MetaBox-v1 by a) universal training and testing interfaces, b) coding examples of different MetaBBO types (at least one for each type), and c) common optimization problem types (at least one for each optimization domain). We hope this primary contribution could be recognized. Second,  to the best of our knowledge, MetaBBO-NE works are very limited, exsiting works such as LES, LGA have been included in MetaBox-v2’s baseline repo.  Third, MetaBBO focus on BBO problems, which are normally continuous optimization problems. Considering this positioning context, LLM-based MetaBBO approaches are still limited, most of them focus on combinatorial domain. At last, as we promised in the end of our paper, given the MetaBox-v2’s universal interfaces, extendability and compatibility toward external resources, we will continuously pay attention to cutting-edge methods and testsuites to maintain an active ecosystem for MetaBBO. For example, LLM-based optimization toolbox LLM4AD [1] and representative testsuites such as pymoo [2] have been writen in our update plans, we hope you can continue to pay attention to our future updates!
>
> [1] Liu F, Zhang R, Xie Z, et al. Llm4ad: A platform for algorithm design with large language model[J]. arXiv preprint arXiv:2412.17287, 2024.
>
> [2] Blank J, Deb K. Pymoo: Multi-objective optimization in python[J]. Ieee access, 2020, 8: 89497-89509.
>
> ### [W2: Testing Workflow]
>
> We understand your concern on the maintainence reliability. We would like to first clarify that we have included manual testing workflow behind version update of MetaBox-v2. There are two branches in our github repository: main and dev. Since we are receiving a lot of user feedback and maintaining monthly updates, we will first update in dev. This is divided into two sub-steps. First, each modification will be tested by our developping member to ensure correctness. Second, dev will be released in advance within a testee group, of which a group of high-level developers will undergo a week of operability checking. Once debug and checking finish, the update in dev will be pushed to main branch, which will be further checked and commited by our major developpers.
>
> Following your kind suggestion, in the next version update, we will include automatic UT and CI framework to systamatically manage and matain resources in MetaBox-v2.
>
> We hope the above responses could address your concerns. Any further comments are welcomed.

---

### Official Review · Reviewer_qbvQ · 2025-07-03

**Rating:** 5
**Confidence:** 3

**Summary:**

Meta Black-Box Optimization (MetaBBO) focuses on learning algorithms that solve black-box optimization problems, i.e problems where gradient information is unavailable.
MetaBBO is an active and largely empirical driven research area. This paper introduces a new benchmarking suite, MetaBox-v2, which extends the capabilities of the existing MetaBox-v1 framework to foster research in MetaBBO. Key improvements include broader support for different algorithmic approaches, the inclusion of more diverse black-box optimization types (e.g., multi-objective problems), and technical enhancements aimed at accelerating both training and evaluation.

**Additional Feedback:**

- The LaTeX template appears to be incorrectly configured. For example, line numbers are missing.

- It might be helpful to map the notation in Section 2 to the elements shown in Figure 2.

- Are the performances across training and test instances normalized when computing the Anti-NFL score in Equation 2?

**Dataset Code Accessibility:**

Yes

**Dataset Code Comments:**

The code appears to be readily accessible:

- The README includes clear installation instructions and example files.

- The repository seems to be properly versioned.

As a suggestion for future improvements, I would recommend adding unit tests and continuous integration (CI) to help ensure the stability and reliability of the library.

**Ethical Considerations:**

No, there are no or only very minor ethics concerns

**Final Justification:**

The authors for addressed my concerns during the rebuttal and I recommend acceptance.

**Limitations Weaknesses:**

- The notation in the paper is somewhat sloppy, and several acronyms and terms are undefined, making it difficult to follow in places. For example:

    - Perf() is not defined in Equation 2.

    - "NFL" is used but not defined.

    - The paper does not precisely define several symbols. For example, it is unclear what the design w_i^t​ refers to and how it is used in the context of algorithm A.


- The section on parallelization could be improved for clarity. For example, the concept of a vectorized environment may not be immediately intuitive to all readers. Additionally, the paper would benefit from a more detailed explanation of the different modes available in Ray, especially for those less familiar with the library.

**Strengths Contributions:**

-  The paper provides a clear motivation for the improvements made over the previous version of the benchmarking suite.

  -  It presents compelling evidence that MetaBox-v2 achieves significantly better performance in terms of runtime compared to MetaBox-v1.

 -   The empirical evaluation appears thorough, incorporating a variety of baseline approaches.

---

> ### Author Rebuttal · Authors · 2025-07-30
>
> We would like to express our deepest gratitude to the reviewer for recogonizing our MetaBox-v2 as a comprehensive benchmark with flexibility, scalability and extensibility. For your remaining concerns, we hope the following point-to-point responses could address them.
>
> ### [W1: Notation Clarity]
>
> - We apologize for the missing definition of $Perf()$. Given a BBO optimizer or MetaBBO baseline $\mathcal{A}$ and a test problem set $D$ (w.o.l.g., all instances are minimization problems),  $Perf(\mathcal{A},D)$ measures normalized optimization performance of $\mathcal{A}$ averaged over all instances in $D$:
>
> $$
> Perf\left(\mathcal{A}, \mathbb{D}\right) = 1 - \frac{1}{|\mathbb{D}|}\frac{1}{51}\sum\_{p\in\mathbb{D}}\sum\_{i=1}^{51}\frac{y^*\_{p,i} - y^\*\_p}{y^{(0)}\_{p,i} - y^\*\_p}
> $$
>
>
> Here we run $\mathcal{A}$ to optimize each instance in $D$ for independent 51 runs. $y^\*\_{p,i}$ denotes the  best objective value found in $i$-th run of instance $p$. $y^{(0)}\_{p,i}$ denotes the best objective value within the initialized population in $i$-th run of instance $p$. $y^*\_p$ denotes the optimal objective of instance $p$ itself. We will add this elaboration into the revised paper.
>
> - “NFL” is the abbreviation of **n**o-**f**ree-**l**unch. We will add "no-free-lunch (NFL)" to the original paper to avoid confusion for readers.
> - Let us explain more on the algorithm design $w\_i^t$ for you. Considering that MetaBBO provides a universal concept framework for diverse algorithm design tasks such as algorithm selection, algorithm configuration etc., $w\_i^t$ is hence a flexible and abstract concept to represent specific design choice under the given context. For example, in algorithm selection like MetaBBO,  $w\_i^t$ denotes the selected algorithm to engage the $t$-th step optimization. We will refine the above explanation into the revised paper.
>
> ### [W2: Parallel and Distributed Acceleration]
>
> - Let us explain for you the “vectorized environment” used in MetaBox’s training paradigm. A naive serial training process looks like the pesudocode below:
>
> ```
> 1. for each problem p
> 2.      for each for each independent run
> 3.             MetaBox.train(A, p)
> ```
>
> Here for each instance $p$ in the training set, we want the meta-level policy to simultaneously interplay with a batch of  low-level optimization processes (line 2), so as to stabilize the normal training (for RL-based MetaBBO, such training trick could release the variance of training signals).  The vectorized environment we proposed leverages multi-processing to help MetaBBO algorithm A  interplay with this batch of low-level optimization process in parallel, which could significantly improve the training efficiency.
>
> - The Ray used in testing paradigm of MetaBox-v2 provides more flexibility (i.e., the four different modes) for users to select suitable parallel scale in light of their concrete hardware settings. Let us first show you a naive testing workflow if we do not use Ray-based acceleration as below:
>
> ```
> 1. for each probelm p
> 2.       for each baseline b
> 3.                      for each independent run r
> 4.                              MetaBox.evaluate(p, b, r)
> ```
>
> Such a nested 3-layer loop suffers from incalculable computational time. Suppose we have N problem in the testing set, B baselines to test, and R independent runs to ensure the statistical robustness, then our model-1 to mode-4 provides various granularities catering to users’ specific computational setting:
>
> ***Mode-1***：N cpu cores are used to distribute the N testing problem instances, on each core, line2-line3 are executed as two-layer loop.
>
> ***Mode-2***:  R cpu cores are used to distribute the R independent runs, while on each core, line1-line2 are executed as two-layer loop.
>
> ***Mode-3:***  NxB cpu cores are used to distribute the N problem instances and B baselines, while on each core, line3 are executed as one-layer loop.
>
> ***Mode-4:***  NxBxR cpu cores are used to distribute all evaluation tasks, hence there is no loop anymore.
>
> We will refine our paper to make the above content more straightforward and clearer for readers.
>
> ### [Additional Feedback]
>
> - We apologize for the inconvenience caused by us, the line numbers are missing indeed, we think this is because we use the onymous latex template.
> - We appreciate your suggestion to enhance the mapping between Section 2 and Figure 2. Refinement will be revised into the paper once it was accepted.
> - Since the Anti-NFL is computed based on the $Perf()$ metric, and $Perf()$ is the normalized performance of a baseline across various problem instances and independent runs, the Anti-NFL score  is deemed to reflect normalized performance to some extents. However, since we apply exponential operation to obtain the final scores, these scores are not normalized ones.
>
> ### [Dataset Code Comment]
>
> We would like to clarify that currently the CI and UT workflow are executed by our team members manually in the “dev” branch, the “main” branch is the source code after these checks. Following your suggestion, we will include automatic UT framework in the next update.
>
> At last, we thank the reviewer again for your careful reading and comprehensive feedback. If there are further concerns, we welcome your timely feedback!

---

### Official Review · Reviewer_u3ra · 2025-07-05

**Rating:** 5
**Confidence:** 5

**Summary:**

This paper upgrades the original open-source Meta-Black-Box Optimization benchmark MetaBox to MetaBox-v2 by integrating efficient parallelization schemes, custom analysis/visualization tools, and more benchmark suites and baselines spanning single-objective, multi-objective, multi-model and multi-task optimization scenarios. In the experiments, the authors present the benchmark results of BBO and MetaBBO algorithms conducted using the proposed benchmark demonstrating the effectiveness of these upgrades.

**Additional Feedback:**

1. In Section 3.2, the authors introduce 4 parallel testing modes. How should a user choose a proper parallel mode?

2. The calculation of the Anti-NFL Indicator requires the difference between the baseline's performance on different test suites. However, these test suites may contain a different number of instances with different value scales. How can we ensure their performance is comparable?

3. In Figure 4, the training parallel mechanism shows different acceleration rates across baselines. What are the key factors influencing these acceleration rates?

**Dataset Code Accessibility:**

Yes

**Dataset Code Comments:**

The authors provide the codes of the proposed benchmark and experiments in the paper on Github, the example metadata is available on huggingface.

**Ethical Considerations:**

No, there are no or only very minor ethics concerns

**Limitations Weaknesses:**

1. The distribution of problem instances and baselines is unbalanced across problem classes. SOO and MOO test suites contain more instances than other test suites, and MetaBBO-RL methods outnumber the other three paradigms.

2. More detailed introduction of the proposed evaluation metrics would benefit this paper. For example, the meaning of ‘Perf()’ is undefined. How these metrics reflect the learning efficiency and generalization ability of the baselines deserves more discussion.

**Strengths Contributions:**

1. The paper is well written. The authors comprehensively describe the benchmarking problems and how the benchmark is composed. The documentation provides sufficient detail and guidance for users to leverage the package for developing and validating their own algorithms.

2. The proposed benchmark incorporates numerous problem instances and baselines for comparison, covering diverse optimization classes and both realistic / synthetic scenarios.

3. The efficient parallelization scheme significantly accelerates the training and testing of MetaBBO methods, potentially accelerating the development of MetaBBO algorithms.

4. This paper proposes novel and comprehensive evaluation metrics for methods in this domain. Coupled with extensive and extensible analysis/visualization interfaces, these enable straightforward comparison and analysis of MetaBBO methods.

---

> ### Author Rebuttal · Authors · 2025-07-30
>
> Dear reviewer, we appreciate you for  recognition of our work in terms of clear writing, detailed documentation, diversity and efficiency in benchmarking evaluation. For your remaining concerns, we provide following point-to-point responses.
>
> ## Limitations Weaknesses:
>
> > 1. The distribution of problem instances and baselines is unbalanced across problem classes. SOO and MOO test suites contain more instances than other test suites, and MetaBBO-RL methods outnumber the other three paradigms.
> >
>
> Firstly, we **admit** that SOO/MOO test suites and MetaBBO-RL methods **are more numerous than others**. This is because most existing MetaBBO approaches focus on using RL agents to control low-level optimizers for SOO and MOO problems.
>
> Secondly, MetaBox-v2 is an open-source, extendable platform. We will **continuously** integrate emerging algorithms and problem **sets**, with an open-source ecosystem **that welcomes** community contributions. In the future, more algorithms from all four paradigms and diverse problem classes will be incorporated into MetaBox-v2 to establish it as a balanced platform.
>
> > 2. More detailed introduction of the proposed evaluation metrics would benefit this paper. For example, the meaning of ‘Perf()’ is undefined. How these metrics reflect the learning efficiency and generalization ability of the baselines deserves more discussion.
> >
>
> ***W2Q1:*** We apologize for the missing definition of $Perf()$. Given a BBO optimizer or MetaBBO baseline $\mathcal{A}$ and a test problem set $D$ (w.o.l.g., all instances are minimization problems),  $Perf(\mathcal{A},D)$ measures normalized optimization performance of $\mathcal{A}$ averaged over all instances in $D$:
>
> $$
> Perf\left(\mathcal{A}, \mathbb{D}\right) = 1 - \frac{1}{|\mathbb{D}|}\frac{1}{51}\sum\_{p\in\mathbb{D}}\sum\_{i=1}^{51}\frac{y^*\_{p,i} - y^\*\_p}{y^{(0)}\_{p,i} - y^\*\_p}
> $$
>
> Here we run $\mathcal{A}$ to optimize each instance in $D$ for independent 51 runs. $y^*\_{p,i}$ denotes the  best objective value found in $i$-th run of instance $p$. $y^{(0)}\_{p,i}$ denotes the best objective value within the initialized population in $i$-th run of instance $p$. $y^\*\_p$ denotes the optimal objective of instance $p$ itself.
>
> ***W2Q2:*** Regarding the Anti-NFL metric: A low value indicates significant performance degradation on unseen testsuites apart from the training set, indicating weak generalization ability.  A higher value means the model retains a similar or even better performance level on unseen problems, overcoming the limitations implied by the no-free-lunch theorem.
>
> ## Additional Feedback
>
> > 1. In Section 3.2, the authors introduce 4 parallel testing modes. How should a user choose a proper parallel mode?
> >
>
> ***AD1Q1:*** The four parallel modes are design for users with different hardware capabilities.
>
> 1. Parallel mode-1 partially enables distributed solving across problem instances by distributing batched problem instances. It has the lowest degree of parallelism. Users with limited hardware resources can use this mode with customized batch sizes to accelerate testing.
> 2. Parallel mode-2 disables problem instance distribution and enables parallel execution of independent test runs. It initiates one process per test run (e.g., 51 processes for 51 runs). This mode requires hardware with over 51 CPU cores.
> 3. Parallel mode-3 enables problem instance distribution and disables test run parallel for instance-wise solving. It launches one process per problem instance. Testing on suites with fewer instances (e.g., BBOB) requires fewer resources, while intensive suites (e.g., HPO-B) require high-performance hardware.
> 4. Parallel mode-4 runs all baselines, problem instances, and test runs concurrently. It has the highest degree of parallelism, achieving the shortest run time while demanding the greatest hardware resources.
>
> > 2. The calculation of the Anti-NFL Indicator requires the difference between the baseline's performance on different test suites. However, these test suites may contain a different number of instances with different value scales. How can we ensure their performance is comparable?
> >
>
> ***AD2Q1:*** As mentioned in the response to Weaknesses 2, Perf is calculated as the averaged normalized best cost across all problems and runs. Averaging model performance over all problems in the testsuites enables comparison of testsuites with different numbers of instances, while using the initial best cost as the normalizer brings value scales of problems in different testsuites to the same level. Therefore, the performance of baselines across different testsuites is comparable.
>
> > 3. In Figure 4, the training parallel mechanism shows different acceleration rates across baselines. What are the key factors influencing these acceleration rates?
> >
>
> ***AD3Q1:*** The key factors influencing the acceleration rates are learning paradigms and how meta-level agents interact with low-level optimizers. For learning paradigms, MetaBBO-NE approaches such as LES learn the policy using nested evolutionary optimization and MetaBBO-ICL approaches such as OPRO requires expensive LLM calling, they hold the lowest efficiency. For interaction modes, MetaBBO-RL approach DEDDQN selects mutation operators per individual, requiring tens of thousands of optimization steps and reducing efficiency. In contrast, another MetaBBO-RL approach RLDAS periodically selects candidate algorithms, taking many fewer steps per optimization process and achieving the highest efficiency.
>
> Thank you again for your professional advice. We hope our reply can reduce your confusion. We welcome any possible questions in the future and look forward to your timely reply!

---

> > ### Comment · Reviewer_u3ra · 2025-08-06
> >
> > Thanks a lot for the reply. The authors’ response have addressed my concerns, as well as supporting my score.

---

### Decision · Program_Chairs · 2025-09-18

**Decision:**

Accept (poster)

**Comment:**

All reviewers voted Accept based on grounds of (I quoted) (1) the paper comprehensively describedetail and guidance for users to leverage the package for developing and validating their own algorithms, (2) the proposed benchmark incorporates numerous problem instances and baselines for comparison, covering diverse optimization classes and both realistic / synthetic scenarios, (3) the proposed benchmark updates a popular benchmark to involve parallelization schemes, custom analysis/visualization tools, and more benchmark suites and baselines spanning single-objective, multi-objective, multi-model and multi-task optimization scenarios, (4) this paper proposes novel and comprehensive evaluation metrics for methods in this domain, (5) the runtime is significantly improved. I mostly agree and make my recommendation accordingly.

===== FINAL UPDATE FROM DB Track PCs ====

The final decision for this paper has been taken by the program chairs after consultation with the SACs. All Senior Area Chairs have ranked papers according to the feedback from the AC during the review process. We decided to leave the original meta-review to reflect the opinion of the AC in light of the initial discussions with reviewers and SAC.